# Effect of Pelvic Floor Muscle Training Using Pressure Biofeedback on Pelvic Floor Muscle Contraction and Trunk Muscle Activity in Sitting in Healthy Women

**DOI:** 10.3390/healthcare10030570

**Published:** 2022-03-18

**Authors:** Min-Joo Ko, Min-Suk Koo, Eun-Joo Jung, Won-Jeong Jeong, Jae-Seop Oh

**Affiliations:** 1Center of Exercise, HSD Engine, 67 Gongdan-ro, Seongsan-gu, Changwon-si 51561, Gyeongsangnam-do, Korea; 99040456@hanmail.net; 2Department of Korean Medicine Rehabilitation, Dang Dang Korean Medicine Center, 674 Wonidae-ro, Seongsan-gu, Changwon-si 51495, Gyeongsangnam-do, Korea; koosi2015@gmail.com; 3Center of FM Pilates, 35 Centum dong-ro, Haeundae-gu, Busan 48059, Korea; cool1118z@hanmail.net; 4Department of Physical Therapy, Gimhaebokum Hospital, 33 Hwalcheon-ro, Gimhae-si 50923, Gyeongsangnam-do, Korea; wonjeong-pt@naver.com; 5Department of Physical Therapy, College of Healthcare Medical Science and Engineering, Inje University, 607 Inje-ro, Gimhae-si 50834, Gyeongsangnam-do, Korea

**Keywords:** pelvic floor muscle training, pressure biofeedback, trunk muscles, verbal feedback

## Abstract

Pelvic floor muscle training (PFMT) has been recommended as the first choice as one of the effective methods for preventing and improving urinary incontinence (UI). We aimed to determine whether pressure biofeedback unit training (PBUT) improves short term and retention performance of pelvic floor muscle contraction. The muscle activities of the external oblique (EO), transversus/internal oblique (TrA/IO), multifidus (MF) and the bladder base displacement were measured in the verbal feedback group (n = 10) and PBU group (n = 10) three times (baseline, post-training, and at the 1-week follow-up). Surface electromyographic activity was recorded from the EO, TrA/IO, and MF muscles. The bladder base displacement was measured using ultrasound. The results were analyzed using two way mixed ANOVA. The bladder base displacement may have elevated more in the PBU group than in the verbal feedback group due to decreased TrA/IO activity. These findings indicate that PBUT is a better method than verbal feedback training.

## 1. Introduction

Urinary incontinence (UI) is defined as any complaint of involuntary leakage of urine [1]. UI has major effects on a woman’s quality of life and health, as it results in mental-social disorders, such as depression, lack of self-confidence, limitations in daily life activities, and feelings of worthlessness [1,2]. The main UI treatments are surgical and conservative [3]. Conservative treatments are more common due to the risk of recurrence of stress UI, which ranges from 10 to 40% after surgical intervention, and the high cost and potential for side effects of surgery [4]. Among the conservative methods for treating UI, pelvic floor muscle training (PFMT) has been recommended as the first choice method to prevent and improve UI [3].

Teaching clients how to contract their pelvic floor muscles (PFMs) is one of the most difficult tasks for a physical therapist [4]. Verbal instruction is commonly used for PFMT in a clinical setting because it is easy, safe, and cost-effective [5]. Previous studies have suggested that biofeedback (vaginal cones, ultrasound, and electrical stimulation) restores continence better than verbal only instruction during PFMT [6,7,8]. However, these methods are invasive, unfavorable, and require expensive equipment and practical experience.

Pressure biofeedback unit (PBU) provides a cost-effective, readily available clinical tool to objectively assess abdominal muscle function, including transverse abdominis (TrA) activation, during the abdominal drawing-in maneuver (ADIM) [9,10]. The device is placed between the perineal region and a solid chair, on which the patient sits, to assess PFM function, and a pressure gauge that detects movement of the PFMs via changes in pressure applied to an air-filled bladder. Correct performance of PFMT results in decreased pressure, indicating the elevation of the pelvic floor. Individuals with high intra-abdominal pressure (IAP) and insufficient PFM activity can increase pressure by a descent of the pelvic floor. Thus, changes in pressure are used to determine PFM activity indirectly.

Previous studies have suggested differences in PFM activity in healthy women depending on the lumbar curve [11,12]; PFM activity is greater when sitting in lumbar lordosis than when sitting in a slumped position [11]. Trunk muscle activity is also affected by lumbo-pelvic posture. Trunk muscle activity decreases in lumbar flexion [11]. Capson et al. [12] examined the activation of the PFMs in neutral, exaggerated, and diminished lumbar lordosis postures and found the highest activation of the PFMs when the patient assumed a neutral position. However, whether the PFMs and trunk muscles are affected by a neutral sitting posture with pressure biofeedback training has not been studied.

Therefore, the objectives of this study were to examine whether pressure biofeedback is effective for teaching pelvic floor muscle contraction (PFMC) using ultrasound; to confirm with pressure biofeedback that TrA/IO, external oblique (EO), and multifidus (MF) muscle activity results in PFM contraction; and to determine whether pressure biofeedback training improves the short-term and retention performance of PFMC.

## 2. Materials and Methods

### 2.1. Participants

Twenty healthy women (age 18–65 years), who had no history of stress urinary incontinence, were not pregnant or had not given birth in the previous 12 months, and were in good general health, were invited to participate. Participants were recruited from recruitment advertisements posted on the FM pilates center bulletin board. The women were randomly allocated to one of two groups by selecting a sealed envelope containing either the number 1 or 2. Group 1 (verbal feedback) contained 10 women (mean ± standard deviation [SD], 36.4 ± 3.5 years; 162.8 ± 4.3 cm; 50.3 ± 4.6 kg) and group 2 (pressure biofeedback) contained 10 women (mean ± SD, 31.7 ± 3.9 years; 162.5 ± 4.1 cm; 51 ± 4.1 kg). Exclusion criteria included being pregnant or less than 3 months postpartum, having systemic neuromuscular disease, previous surgery or intensive PFMT, or severe low back pain or pelvic pain. All subjects provided informed consent. This study was approved by the institutional review board of Inje University in the Republic of Korea (No. INJE 2020-01-044-001).

The sample size was estimated using G*Power 3.1.9.7 for Windows (G*Power©, University of Dusseldorf, Dusseldorf, Germany). A power analysis determined that at least a total 16 subjects were required to obtain a power of 0.8 at an α level of 0.05. This analysis was based on an estimated effect size derived from the previous literature [13]. Allowing for a dropout rate, we decided to recruit at least 10 participants per group (a total of 20 subjects) in order to provide sufficient power to detect significant group differences.

### 2.2. Procedure and Interventions

Subjects were randomly assigned to 2 groups using a table of random numbers created by online software (www.randomization.com, accessed on 23 October 2021): a verbal feedback group, who received only verbal feedback training; a pressure biofeedback group, who received pelvic floor muscle training with pressure biofeedback. The verbal feedback training was performed as follows: The subject was seated in a lumbo-pelvic upright position (anterior rotation of pelvis, lumbar lordosis and relaxation of the thorax). The subjects were instructed to contract their pelvic floor muscle by the description “squeeze and lift your pelvic floor muscles as if trying to stop the flow of urine” (Figure 1). For the pressure biofeedback training, the subject was seated in the lumbo-pelvic upright position. The pressure biofeedback unit was placed between the pad of the chair and the subject’s perineal region to monitor PFM contraction. The subject was instructed to squeeze and lift pelvic floor muscle maximally and hold the position. The subjects were asked to maintain the changed pressure by visual feedback from a digital pressure gauge during pelvic floor muscle contraction (Figure 1). Approximately 5 min of biofeedback was provided to optimize the performance of the PMFC. Five minutes was selected based on research by Dietz et al. [14] in the use of biofeedback for the instruction of pelvic floor contractions and clinical feasibility. Each subject was held for 3 s with a resting break of 10 s between each contraction. All outcome measures including bladder base displacement and trunk muscle activities (TrA/IO, EO, MF) were carried out 3 times during the study: before the training period, 3 min after the training period and 1 week after the training period.

### 2.3. Measurements

Displacement of the bladder base was assessed by transabdominal ultrasound imaging using a portable ultrasonography device (SONON 300L, Healcerison, Seoul, Korea) with a curved transducer (3.5 MHz). A bladder-filling protocol was implemented before testing to ensure that subjects had sufficient fluid in their bladders to allow clear imaging. This protocol involved subjects consuming 450–500 mL of water in a 1 h period, which was completed 30 min before testing time. The assessment of PFM thickness was conducted by ultrasound (Dornier Medtech 7.5-MHz straight linear array transducer). The ultrasound transducer was placed in a transverse orientation, across the middle of the abdomen, immediately superior to the pubic symphysis for transverse plane transabdominal ultrasound imaging of the bladder base. The angle of the transducer was manipulated until it was approximately 60° from the vertical and aimed towards the bladder base (Figure 2). The angle of the ultrasound transducer was adjusted until there was a clear image of the bladder and midline pelvic floor structures (urethra, perineal body, and rectum). An “X” mark was placed on the bladder base at rest. The subject was then asked to perform a voluntary PFMC, and the image was captured at the point of maximal displacement, and this point was marked with an “X”. Displacement of the pelvic floor elevation from the resting position at the end of each contraction was measured. Displacement was measured as the distance between the two “X” points in millimeters (mm).

A Trigno wireless EMG system (Delsys, Inc., Boston, MA, USA) was used to assess the electromyographic (EMG) activity of the bilateral TrA/IO, EO, and MF muscles. The sampling rate was 1000 Hz, with a 20–450 Hz bandpass filter. All raw EMG data were converted into root mean square data for analysis. The electrodes for the TrA/IO, EO, and MF muscles were placed according to Criswell [15]. The site for each electrode was shaved and then cleaned with cotton and alcohol to reduce skin impedance. To normalize the EMG activity of the IO, EO, and MF muscles, the maximum voluntary isometric contraction (MVIC) of the muscles was measured using maneuvers suggested previously [16].

The MVIC trials for each muscle were performed twice with a rest period of 1 min between trials, and the mean EMG value of the middle 3 s of three trials was used to normalize the value for each muscle. All EMG data during pelvic floor muscle contraction are expressed as percentages of MVIC (%MVIC).

### 2.4. Statistical Analysis

The independent *t*-test was used to evaluate group differences at baseline. The one-sample Kolmogorov–Smirnov test was employed to ensure a normal distribution of the measurement data. A two-factor, mixed-model analysis of variance (ANOVA) (2 groups × 3 time-points) was used to compare outcome measures between groups over time. This analysis was repeated for muscle activities (TrA/IO, EO, and MF) and bladder base displacement. If a significant interaction was found, independent *t*-tests were used to compare the between-group differences for the bladder base displacement and muscle activities at each of three time-points (pre-, post-, and retention-testing). Additionally, paired *t*-tests were performed for comparison of those parameters between time points (pre-testing vs. post-testing, pre-testing vs. retention-testing) within each of the groups. All statistical analyses were conducted with SPSS Statistics version 18 (SPSS Inc., Chicago, IL, USA). A *p*-value < 0.05 was considered significant.

## 3. Results

The demographic characteristics, muscle activities (TrA/IO, EO, and MF), and bladder base displacement were comparable between the groups at baseline. All subjects completed the post test and retention test.

The ANOVA evaluating bladder base displacement between groups across the three times revealed a significant group × time interaction (F = 3.54, *p* = 0.04). Within-group post hoc testing demonstrated that the pressure biofeedback group exhibited a significant elevation in the bladder base from pre-test to post-test (t = 5.783, *p* < 0.001) and from pre-test to retention-test (t = 5.261, *p* = 0.001). The verbal feedback group exhibited a significant elevation in the bladder base from pre-test to post-test (t = 3.199, *p* = 0.013) and from pre-test to retention-test (t = 3.049, *p* = 0.016). Between-group post hoc testing revealed that the bladder base was elevated in the pressure biofeedback group compared to the verbal feedback group on the post-test (t = 3.153, *p* = 0.006) and retention-test (t = 2.933, *p* = 0.014) (Table 1).

The ANOVA evaluating the activities of the TrA/IO muscles between groups across the three time points revealed a significant group × time interaction (Rt. TrA/IO; F = 4.623, *p* = 0.017, Lt. TrA/IO; F = 3.934, *p* = 0.03). Within-group post hoc testing showed that TrA/IO activity in the pressure biofeedback group decreased significantly from pre-test to post-test (Rt. TrA/IO; t = −3.436, *p* = 0.007, Lt. TrA/IO; t = −2.686, *p* = 0.025) and from pre-test to retention-test (Rt. TrA/IO; t = −3.492, *p* = 0.007, Lt. TrA/IO; t = −2.864, *p* = 0.019). However, the verbal feedback group demonstrated no significant difference across the three time points. Between-group post hoc testing revealed that TrA/IO muscle activity was lower in the pressure biofeedback group than in the verbal feedback group on the post test (Rt. TrA/IO; t = 2.240, *p* = 0.04, Lt. TrA/IO; t = 3.329, *p* = 0.004) and retention test (Rt. TrA/IO; t = 2.617, *p* = 0.022, Lt. TrA/IO; t = 3.165, *p* = 0.006) (Table 2).

The ANOVA evaluating the activity of the EO muscle (Rt. EO; F = 2.594, *p* > 0.05, Lt EO; F = 2.896, *p* > 0.05) and MF muscle (Rt. MF; F = 0.947, *p* > 0.05, Lt. MF; F = 0.1, *p* > 0.05) between groups across the three time points revealed no significant group × time interaction.

## 4. Discussion

Both verbal feedback and pressure biofeedback training elevated the bladder base. Improvements in both groups were maintained at the 1-week follow-up. The mean post-training change in the bladder base in the verbal feedback group was 1.97 mm, whereas the change for the pressure biofeedback group was 4.48 mm. Despite these improvements, the increase in the bladder base was greater in the pressure biofeedback group at the 1-week follow-up. Pressure biofeedback training significantly decreased TrA/IO muscle activity, which was maintained at the 1-week follow-up, but verbal feedback training did not at either the post-training or 1-week follow-up. The decrease in TrA/IO muscle activity was significantly greater in the pressure biofeedback group than in the verbal feedback group and it was the same at the 1-week follow-up, suggesting that reduced TrA/IO activity contributed to improving selective contraction of the PFMs. However, the EO and MF muscle activities of the groups across the three time-points revealed no significant group × time interaction.

Both verbal feedback and pressure biofeedback training elevated the bladder base, and the improvements in both groups were maintained at the 1-week follow-up. Augmented feedback might contribute to enhancing motor learning. There are two broad types of feedback, intrinsic (via sensory systems within the body) and extrinsic (augmented), which is supplementary feedback about the movement in addition to what is provided through intrinsic feedback. Typical types of augmented feedback used to improve motor performance include auditory, visual, verbal, and somatosensory. Augmented sensory feedback facilitates muscle activation during the early stages of learning [17]. Furthermore, motor learning is characterized by a lasting increase in performance that is assessable on short- and long-term retention tests when augmented feedback is withdrawn [18]. Therefore, augmented feedback, including verbal and pressure biofeedback training, may elevate the bladder base for 1 week by automatization, which is the final stage of motor learning.

Despite the improvements realized in both groups, the increase at the bladder base was greater in the pressure biofeedback group, suggesting that the pressure biofeedback unit is useful as a biofeedback tool. There are several reasons why pressure biofeedback may be more useful than verbal feedback for improving PFM motor learning. The first is that pressure biofeedback, in addition to tactile and visual feedback, provides multimodal learning compared to verbal feedback. Previous studies have suggested that the threshold of neural activation is reached earlier by multimodal learning than by unimodal learning [19]. Particularly in the early attention-demanding learning phase, concurrent augmented feedback may help the novice to understand the new structure of the movement faster and prevent cognitive overload, which accelerates the learning process [18]. Pressure biofeedback might have a greater effect on motor learning because of concurrent feedback compared to verbal feedback. A second reason is that pressure biofeedback training, which is externally focused feedback, facilitates automatic control of movement as well as movement efficiency compared to verbal feedback training, which is internally focused feedback. Verbal feedback is focused on the performer’s body movements (internal focus), while pressure biofeedback is focused on the movement effect (external focus) [20]. Instructions or feedback promoting an external focus enhance learning, compared to those inducing an internal focus. Adopting an external focus has been shown to facilitate automatic movement control [20] as well as movement efficiency [21]. Wulf and Lewthwaite [22] proposed that internal focus instruction promotes self-awareness and excessive concerns about own movements. Such a phenomenon was described as “micro-choking”. Therefore, pressure biofeedback training (external focus) is a more useful method to train the PFMs than verbal feedback training (internal focus) because it increases the automaticity of movement control and movement efficiency.

The reason why pressure biofeedback training is more useful than verbal feedback training to elevate the bladder base is explained by the decrease in TrA/IO muscle activity. The reason for the decrease in TrA/IO muscle activity in the pressure biofeedback training group may be attributable to surround inhibition (SI). SI within the primary cortex is expressed as a selective reduction in motor-evoked-potential amplitudes in surrounding muscles before and during contraction of the target muscle compared to a resting condition [23]. Furthermore, Kuhn et al. [24] demonstrated an efficient way to modulate SI by changing the attentional focus in healthy subjects; they found that better motor performance was associated with an external focus. In light of these previous studies, pressure biofeedback training should inhibit the surrounding TrA/IO muscles by the SI mechanism along with elevation of the bladder base. In addition, the decrease in TrA/IO muscle activity could reduce IAP, which would contribute to further elevation of the bladder base as a result of resistance to PFM elevation. However, no significant change in TrA/IO muscle activity occurred in the verbal feedback training group, possibly because those muscles may not be able to fully recruit all motor units to change the muscle activation patterns during maximal force production.

Our results confirm that the EO and MF muscles did not have a predictable pattern of activation in response to the PFM biofeedback training. Our EMG findings are consistent with those of Neumann and Gill [25] suggested that strong PFMC results in strong and simultaneous recruitment of the TrA and IO muscles but not of the EO muscle in asymptomatic subjects. The activation levels of the EO and MF muscles were sufficiently high (approximately 20% MVIC) to suggest that they may play a role in posture during PFMC. O’Sullivan et al. [26] reported that co-activation of trunk muscles, including the EO and MF muscles, increases spinal stability and is necessary to maintain sitting in the lumbo-pelvic position. These findings suggest that the EO and MF muscles may act more as postural muscles to maintain the lumbo-pelvic sitting posture than as co-activators with the PFMs during pelvic floor training.

A possible limitation of the study is that our subjects were urinary continent based on self-report. It is not possible to rule out that some participants had bladder neck incompetence or other pathologies, as urodynamic studies were not performed. Another area of concern is that our sample was sufficiently small and did not include incontinent women. Finally, TA ultrasound may be criticized because of the lack of a fixed bony landmark as a reference point. Measures of displacement are expressed only relative to a chosen starting point rather than an anatomical landmark. In this study, similar to others, the distinct edge of the endopelvic fascia in the region of its greatest observed displacement that was observable during the movement was selected for measurement.

## 5. Conclusions

The use of pressure biofeedback for teaching PFMC to subjects without incontinence is a beneficial teaching tool to facilitate consistent PFMC compared to verbal feedback. The effect of pressure biofeedback on retention of PFMC performance was inconclusive in this study. More research is needed to determine whether subjects with UI would also benefit from the use of pressure biofeedback to learn how to contract the PFMs.

## Figures and Tables

**Figure 1 healthcare-10-00570-f001:**
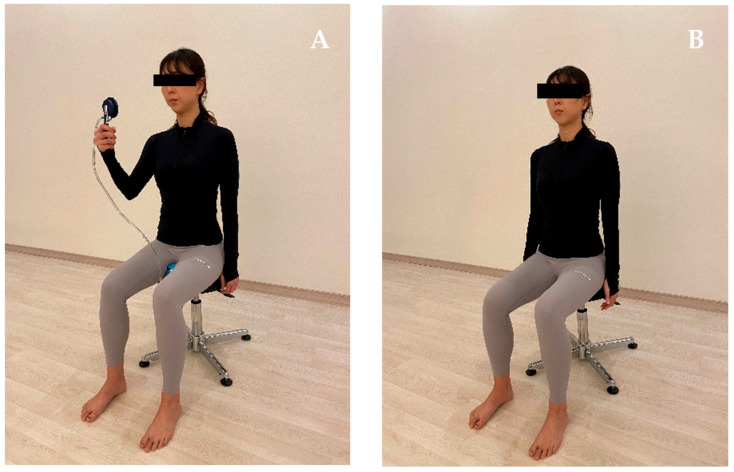
Performing pelvic floor muscle contraction with different feedback methods. (**A**) Pressure biofeedback. (**B**) Verbal feedback.

**Figure 2 healthcare-10-00570-f002:**
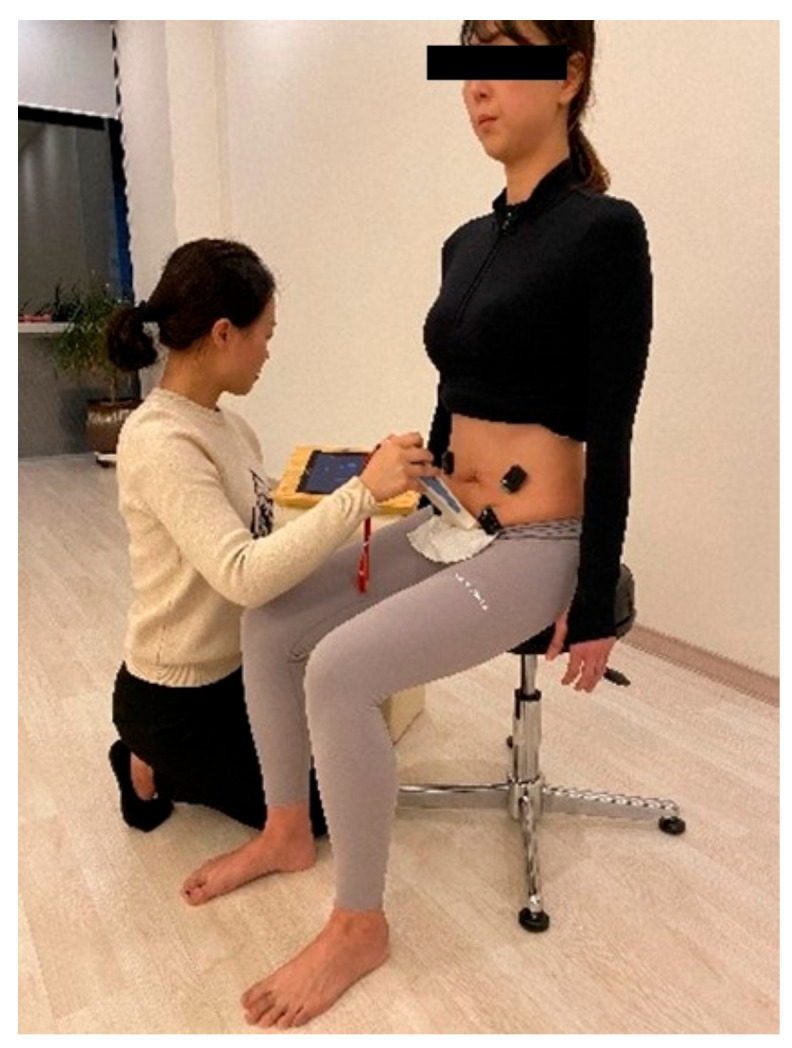
Ultrasound probe and electromyographic (EMG) electrode displacement.

**Table 1 healthcare-10-00570-t001:** Results of bladder base displacement (mm) in response to intervention.

Group	Pre-Testing	Post-Testing (3 min)	Retention-Testing (1 Week)	Time						Group								
				3 min-Pre-Testing			1 Week-Pre-Testing			At Pre-Testing			At Post-Testing			At Retention-Testing		
				95%CI	*p*	Cohen’s d	95%CI	*p*	Cohen’s d	95%CI	*p*	Cohen’s d	95%CI	*p*	Cohen’s d	95%CI	*p*	Cohen’s d
Verbal feedback group	−2.18 ± 4.14	−0.21 ± 3.30	0.23 ± 3.26	−3.34, −0.55	0.013 *	0.526	−4.22, −0.59	0.016 *	0.647	−5.39, 1.53	0.254	0.558	−7.42, −1.45	0.006 **	1.486	−6.04, −0.85	0.014 **	1.382
Pressure biofeedback group	−0.25 ± 2.61	4.23 ± 2.64	3.67 ± 1.33	−6.26, −2.69	0.000 *	1.707	−5.63, −2.20	0.001 *	1.892									

Data are mean ± SD, CI, confidence interval, * Paired *t* test significantly different from pre-testing at *p* < 0.05, ** Independent *t* test significantly different between groups at *p* < 0.05.

**Table 2 healthcare-10-00570-t002:** Muscle activity (%MVIC) in response to intervention.

Group	Pre-Testing	Post-Testing (3 min)	Retention-Testing (1 Week)				*p*											
							Time						Group					
				3 min-Pre-Testing			1 Week-Pre-Testing			at Pre-Testing			at Post-Testing			at Retention-Testing		
				95%CI	*p*	Cohen’s d	95%CI	*p*	Cohen’s d	95%CI	*p*	Cohen’s d	95%CI	*p*	Cohen’s d	95%CI	*p*	Cohen’s d
Verbal feedback group																		
Rt. TrA/IO	43.14 ± 12.81	45.09 ± 16.48 **	47.80 ± 16.83 **	−9.65, 5.76	0.576	0.132	−15.12, 5.81	0.335	0.312	−6.57, 16.22	0.382	0.424	0.76, 27.50	0.040 **	1.055	2.85, 30.62	0.022 **	1.234
Lt. TrA/IO	42.23 ± 10.11	43.51 ± 12.11 **	39.75 ± 7.64 **	−7.29, 4.72	0.635	0.115	−2.53, 7.49	0.286	0.277	−2.16, 15.86	0.127	0.759	6.29, 28.33	0.004 **	1.569	4.29, 21.67	0.006 **	1.491
Rt. EO	40.72 ± 9.64	38.92 ± 9.95	44.07 ± 12.75	−0.16, 3.77	0.067	0.184	−11.29, 4.59	0.359	0.296	−5.87, 12.19	0.469	0.350	−7.70, 11.20	0.700	0.185	−4.06, 17.51	0.205	0.623
Lt. EO	36.95 ± 10.60	36.36 ± 10.64	40.88 ± 8.12	−2.67, 3.84	0.688	0.056	−10.38, 2.53	0.198	0.416	−7.19, 10.69	0.684	0.196	−5.79, 13.16	0.422	0.388	0.63, 15.05	0.069	0.920
Rt. MF	25.41 ± 15.13	21.18 ± 9.53	24.03 ± 12.55	−2.69, 11.14	0.196	0.335	−1.84, 4.59	0.353	0.099	−2.28, 22.02	0.093	0.843	−0.88, 16.21	0.075	0.897	−0.92, 16.25	0.072	0.906
Lt. MF	24.63 ± 14.15	21.17 ± 9.56	24.62 ± 15.01	−1.61, 8.53	0.154	0.287	−3.94, 3.96	0.996	0.001	−6.02, 16.80	0.331	0.472	−4.01, 13.42	0.270	0.535	−7.89, 16.56	0.463	0.354
Pressure biofeedback group																		
Rt. TrA/IO	38.31 ± 9.80	30.96 ± 9.32	31.06 ± 9.21	2.72, 13.23	0.007 *	0.769	2.62, 12.24	0.007 *	0.762									
Lt. TrA/IO	35.38 ± 7.78	26.21 ± 9.83	26.77 ± 9.65	1.25, 14.60	0.025 *	1.034	1.56, 13.31	0.019 *	0.982									
Rt. EO	37.56 ± 8.38	37.17 ± 8.94	37.35 ± 8.38	−1.17, 1.21	0.969	0.045	−1.32, 1.08	0.822	0.025									
Lt. EO	35.20 ± 6.89	32.68 ± 8.17	33.67 ± 7.55	−2.77, 6.66	0.376	0.333	−3.32, 5.45	0.596	0.212									
Rt. MF	15.54 ± 6.74	13.51 ± 7.45	14.99 ± 6.44	−1.25, 4.22	0.250	0.286	−1.72, 2.08	0.839	0.083									
Lt. MF	19.24 ± 7.78	16.47 ± 7.93	20.29 ± 8.61	−0.48, 4.80	0.097	0.353	−6.73, 4.21	0.615	0.128									

Data are mean ± SD, CI, confidence interval, * Paired *t* test significantly different from pre-testing at *p* < 0.05, ** Independent *t* test significantly different between groups at *p* < 0.05.

## Data Availability

Not applicable.

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
