# Peer review of "Effect of Pelvic Floor Muscle Training Using Pressure Biofeedback on Pelvic Floor Muscle Contraction and Trunk Muscle Activity in Sitting in Healthy Women"

_healthcare, 2022, doi:10.3390/healthcare10030570_

Round 1

Reviewer 1 Report

Dear Authors,

This manuscript is an interesting study analyzing new training methods for pelvic floor muscles.

I think it is necessary to revise this manuscript with reference to the following opinions.

Major revision

  1. Please elaborate on the participant's profile and recruiting method. If possible, use a follow chart to illustrate.

  1. Please show me how to calculate the sample size. The number of people in one group is as small as 10. If the sample size is not enough, for example, "Pilot Study" would be good.

  1. Do the two groups have a normal distribution? Also, have you conducted a homoscedastic test? If these two do not hold, then you may need to use a nonparametric test.

Reviewer 2 Report

I have two questions for you.

1. Please check the English translation below again. (Have you submitted an English translation certificate?)

"Wulf and Lewthwaite[21] argued that internal focus instructions or feedback promote a focus on the self, leading to concerns and worries about one’s performance, and, subsequently, “micro-choking” events."

2. Is it possible to modify Table 1 and Table 2 into a table containing p values?
Tables represent the results in an easy-to-understand way instead of explanations, but they are difficult to see.

3. This is a human experiment. Have you obtained an IRB for this study? If not, please explain why.

Round 2

Reviewer 1 Report

The manuscript has been much improved and is in a nice condition now.